# A Study on the Leisure Sports Participation Behavior of the Elderly through Comparative Analyses by Age: Focusing on Leisure Participation Constraints and Price Sensitivity

**DOI:** 10.3390/bs14090803

**Published:** 2024-09-11

**Authors:** Soon-Young Kim

**Affiliations:** Department of Physical Education, Gachon University, 1342, Seongnam-daero, Sujeong-gu, Seongnam-si 13120, Republic of Korea; klpga0166@gachon.ac.kr

**Keywords:** leisure sports participation, older adult, leisure constraints, price-sensitive, super-aged society

## Abstract

Worldwide, interest in healthy living has been increasing as people’s lifespans have lengthened, owing to interest in health and the development of the medical industry. The need for research on healthy lifestyles aided by sports activities for older adults is greater than before. This study aimed to compare and analyze constraints on participation in leisure sports and participation price sensitivity based on age groups in a super-aging society. From 22 May to 10 July 2024, in three community and sports centers, Korean adults over the age of 20 who regularly participated in leisure sports voluntarily responded to a questionnaire. Based on a quantitative research design using a survey with a convenience sampling technique, 305 collected survey responses were analyzed for validity, reliability, and exploratory factor analysis through SPSS 28. Additionally, one-way multivariate analysis of variance and the price sensitivity meter technique allowed us to analyze the differences in leisure participation constraints and price sensitivity among the groups. No statistically significant group differences were found in the health and social factors of leisure sports participation constraints. However, statistically significant differences were found for the cost and time factors. Finally, the price sensitivity meter technique found differences in price sensitivity in participating in leisure sports among the three age groups. This scientific analysis, aiming to expand older adults’ participation in leisure sports, provides objective data for the future.

## 1. Introduction

As the human lifespan is increasing owing to advances in science and medical technology, the size of the adult population aged 65 and older is expected to increase rapidly, especially as the baby boomer generation enters the older adult age group [1]. Aging refers to the increasing proportion of people aged 65 years and above in a country’s population [2]. According to Korea’s 2022 senior citizen statistics, the number of people aged 65 and over was approximately 9 million as of 2022. Statistics Korea projects that the older adult population will reach 20.6% by 2025, making South Korea an super-aging society. By 2030, this population will exceed 30% for the first time, and by 2070, it will increase to 40%. While South Korea’s aging rate is not yet significantly higher than that of other countries, Statistics Korea has noted that the country is aging at a faster rate, especially compared to other OECD countries.

It took only 19 years for Korea to go from an aging society (7%) to a super-aging society (14%), compared to the 24 years of Japan, which is known for its rapid aging rate; this indicates that Korea is among the fastest in the world to become an aging society [3]. According to Statistics Korea’s Life Table data, the average life expectancy in Korea as of 2021 was 83.6 years, more specifically, 80.6 years for men and 86.6 years for women, and it is steadily increasing every year [4]. This rapid aging and increase in life expectancy mean that, unlike developed countries, which have been dealing with aging for a relatively long time, Korea needs to prepare for and act in response to an aging society [5]. Additionally, owing to rapid industrialization and increased life expectancy, the lives of older adults are significantly different from the past. It is necessary to prepare for the increase in the older adult population, especially with regard to older adults’ quality of life and welfare [1].

With the rapidly aging population, the need for physical activity among older adults is increasing. To achieve an active and productive aged society, staying healthy and active throughout the remaining years of life should be prioritized, and vigorous physical activity should be considered a key component of this [5]. Regular physical activity in older adults is necessary as it can provide many physical, mental, and social benefits. However, the current prevalence of moderate physical activity consisting of 30 min or more by age is relatively low, with 11.9% of older adults in Korea practicing physical activity in their 60s, 11.4% in their 70s, and 7.1% in their 80s [6]. All humans age, and aging is a natural process that occurs as the body’s functions decline during the natural aging process [7] and requires a multifaceted approach that includes biological, psychological, and social aspects [8].

In this context, studies analyzing the importance and necessity of physical leisure activities [9,10] have been steadily progressing. However, with life expectancy exceeding 80 years owing to improved medical standards and living conditions and people reaching the age of 100, objective studies on participation in leisure activities in older adults are relatively insufficient. Particularly, most studies on older adult health have been conducted in the field of natural science, and the socio-psychological approach that can activate leisure participation is even more fundamental. After all, before conducting research on older adult health from a medical perspective, the priority is to analyze the reasons why it is relatively difficult for older adults to participate in physical leisure activities for social and psychological reasons and to suggest countermeasures. In addition, previous studies [10,11] on leisure sports participation have adopted age as a major variable, but only one specific age group has been analyzed. It would be meaningful to derive the results of this study only for the elderly through comparative analyses by age groups.

Constraints on leisure participation, which will be addressed as an important concept in this study, are usually due to individual social and psychological factors, which are based on the fact that humans are social animals [12]. As people participate in leisure sports, they experience participation constraints due to various factors, and researchers are working hard to analyze them. The concept of leisure constraints, through the hierarchical three constraints theory, is based on three types of individual internal constraints: (a) the psychological state and attributes of individuals interacting with leisure participation; (b) interpersonal constraints (the results of interpersonal relationships or relationships between individual characteristics); and (c) structural constraints (financial status, lifestyle, climate, and working hours) [13]. This theory aims to understand how leisure constraints might influence individuals’ nonparticipation, but subsequent research has stated that constraints on leisure activities may continue to influence subsequent commitment after participation in an activity [14]. After all, leisure participation constraints have been used as a concept through which to analyze various factors experienced by an individual when participating in leisure activities as a social animal.

Additionally, one concept that will be addressed in detail in this study is cost. In recent years, as the cost of participating in leisure has grown higher than in the past, people’s participation in leisure sports is changing into consumption. In the end, participation in leisure incurs costs, which are an important factor in determining whether people will participate in leisure sports. Cost is a recognized value and a medium between service quality and consumer satisfaction, and it may have a substantial impact on consumers’ future consumption behavior intentions [15]. If so, a question may arise as to what extent leisure participation costs are perceived as appropriate by the public. To answer such questions in relation to rational participation in leisure, this study aims to analyze consumers’ thoughts on subjective prices by applying the price sensitivity meter (PSM) technique [16] introduced by German economist Peter van Westendorp in 1976. Price sensitivity refers to consumers’ response to price [17]; it explains to what extent consumers can accept price changes and provide continuous support.

The PSM technique is widely used in both industry and academia as it is easy to measure and analyze. It has the advantage of being able to measure price sensitivity and analyze it from various angles using only four simple questions. The PSM technique can analyze price sensitivity in various ways, using four surveys to measure consumers’ price change response. Specifically, through this analysis, perceptions of (a) too cheap to question, (b) cheap, (c) expensive, and (d) too expensive to consume [18] are derived using the response distribution based on respondents’ answers. This analysis enables various interpretations by deriving (a) the most ideal price that consumers perceive, (b) the price commonly recognized in the market, (c) the price that is likely to call quality into question owing to too low a price, and (d) the price whose price resistance reaches a limit due to too high a price. This study aims to analyze price sensitivity using the PSM technique to sensibly expand opportunities for older adults to participate in leisure by analyzing the cost of participation in leisure by age in the super-aging era. This study’s results are expected to be used as objective data for rational choices for the popularization of older adults’ leisure participation in the future by analyzing various pieces of information obtained from leisure participants.

This study aims to suggest a plan to revitalize older adults’ participation in leisure as part of a satisfactory life in a super-aging society. Although related research has been continuous [19,20,21], most studies have not considered social factors, and fragmentary analysis has been mainstreamed; thus, such studies have only gained academic significance rather than practical contributions. Therefore, this study compares and analyzes the limiting factors of leisure participation that older adults experience by age. Additionally, it conducts a comparative analysis of the price sensitivity of participation costs based on the practical limitations revealed as a result of the analysis and addresses various social factors. Determining which difficulties older adults are experiencing beyond the concept of age expressed by objective numbers will increase the significance of this study. Ultimately, this study aims to conduct systematic research from a more multidimensional perspective. Specifically, through comparative analysis, it verifies the following two hypotheses:

**Hypothesis** **1.**
*Participation constraints differ depending on the age of participants in leisure sports.*


**Hypothesis** **2.**
*Price sensitivity differs depending on the age of participants in leisure sports.*


## 2. Materials and Methods

### 2.1. Survey Design and Setting

This study aims to analyze the continuous participation of older adults in leisure sports in the current context of a super-aging society with increased life expectancy due to increased interest in health and the development of the medical industry. We compared and analyzed factors related to leisure participation constraints experienced by age; then, we performed data collection to analyze the costs that people incur when participating in leisure sports. Participants in this study were limited to adults over the age of 20 in the Republic of Korea who had been continuously participating in physical leisure activities. Those who were not regularly participating in leisure sports and expressed their intention to participate in the survey were excluded.

### 2.2. Participants and Sample Size

Data collection began on 22 May 2024, after this study obtained the approval of the Institutional Review Board of Gachon University, Republic of Korea, and it ended on 10 July 2024. This study distributed 300 copies of the questionnaire both online and offline and finally collected 286 copies, excluding 14 incomplete questionnaires. This study adopted a quantitative research design using survey method with a convenience sampling technique for data collection and analysis. The respondents completed the questionnaire with a self-assessment method based on voluntary participation and familiarity with the contents of the purpose of this study. The independent variable for comparative analysis was set as the participants’ age and divided into younger adults (ages 18–35), middle-aged adults (ages 36–55), and older adults (aged older than 55) [22]. Table 1 shows detailed demographic information of all survey respondents.

### 2.3. Variables

The constraints on leisure participation dealt with as an important concept in this study are due to important individual social and psychological factors based on the fact that humans are social animals [12]. Crawford and Godbey [13] present three types of individual internal constraints: (a) individual psychological state and attributes that interact with leisure participation; (b) interpersonal constraints (relationship results or personal characteristics); and (c) structural constraints (financial status, lifestyle, climate, and working hours). Leisure participation constraints have been used as a concept through which to analyze various factors experienced by an individual when performing leisure activities as a social animal. This study modified and used the constraints on leisure participation developed in Choi’s [23] study. Finally, this study adopted factors consisting of (a) health (three factors), (b) social (three factors), (c) cost (three factors), and (d) time (three factors).

Next, the price sensitivity meter (PSM) technique has been used reliably thus far to measure psychological sensitivity to leisure participation costs, which have recently attracted attention among participation constraints. The PSM technique has been widely used in both industrial and academic fields, as its form of measurement and analysis is relatively simple and accurate; it allows for various interpretations by measuring price sensitivity with only four simple questions and applying the results. The PSM technique was used in this study to derive the following four results through which to measure the participants’ price sensitivity responses [18,24]: (a) too cheap to question, (b) cheap, (c) expensive, and (d) too expensive to consume. Westendorp’s [17] price sensitivity analysis presents four main results, as shown in Figure 1 below. Each significant price point can be interpreted as follows: (a) point of marginal cheapness (PMC), (b) point of marginal expensiveness (PME), (c) optimal pricing point (OPP), and (d) indifference pricing point (IPP).

### 2.4. Statistical Methods

This study verified the data collected for this research via SPSS version 28.0. First, statistical analysis provided descriptive statistics for the survey respondents, including sociodemographic information (i.e., sex, age, monthly income, career in participating in leisure sports, frequency of participation in leisure sports, and types of participation in leisure sports). Second, this study analyzed the validity of the collected data using exploratory factor analysis (EFA) with the dependent variable (i.e., leisure participation constraints). Third, to test the scale’s internal reliability, this study utilized Cronbach’s alpha coefficients. Fourth, this study conducted a one-way multivariate analysis of variance (MANOVA) to determine differences in the dependent variable between age groups. The MANOVA for comparative analysis is a satisfactory statistical technique that has been used in social science [25,26,27,28,29,30]. Finally, the PSM technique was used to analyze the psychological price sensitivity of leisure sports participants by age.

## 3. Results

### 3.1. Scale Validity and Reliability

This study conducted EFA using principal component analysis (PCA) for the dependent variable (i.e., leisure participation constraints) including four sub-factors: (a) social (three items), (b) cost (three items), (c) time (three items), and (d) health (three items). To verify the factor structure of the leisure participation constraints, the Kaiser–Meyer–Olkin test generated sample adequacy (0.823) [31]. Bartlett’s test of sphericity showed statistical significance (χ^2^ = 3303.576, *df* = 66, *p* < 0.01). As a result, this EFA retained four leisure participation constraints sub-factors (i.e., social, cost, time, and health), accounting for 86.128% of the total variance based on acceptable statistical criterion: (a) greater than one eigenvalue and (b) greater than 0.40 factor structure coefficients. Next, Cronbach’s alpha coefficients of all the sub-factors achieved satisfactory internal consistency (greater than 0.70): cost (*α* = 0.949), health (*α* = 0.845), time (*α* = 0.911), and social (*α* = 0.943) [32,33], as reported in Table 2.

### 3.2. Multivariate Analysis of Variance

In this study, we conducted a one-way MANOVA to test differences in leisure participation constraints by three age groups. First, this study tested the homogeneity of covariance (Box’s *M* = 47.267, *F* = 2.318, *p* < 0.001), showing statistically significant differences among groups (Wilks’ lambda = 0.811, *F* = 8.231, *p* < 0.01, partial *η*^2^ = 0.099). More specifically, comparative analysis showed statistically significant mean differences in the cost and time factors of leisure participation constraints.

After this comparative analysis including more than three age groups, this study performed an additional post hoc analysis to determine which groups specifically revealed statistical differences. First, in terms of the health and social factors of leisure participation constraints, the groups showed no statistically significant mean differences. Next, regarding the cost of leisure participation constraints, the younger age group (G1) showed higher mean scores than the middle age group (G2) and older age group (G3). Finally, regarding the time of leisure participation constraints, the older age group (G3) reported higher mean scores than the rest of the groups (G1 and G2). Detailed results of the MANOVA and post hoc analyses are shown in Table 3 and Table 4.

### 3.3. PSM Technique

In this study, we conducted price sensitivity analysis based on responses to the cost of participating in leisure sports and analyzed (a) PMC, (b) PME, (c) OPP, and (d) IPP by age group. The specific result graphs are shown in Figure 2, Figure 3 and Figure 4.

## 4. Discussion

Rapid transformation into a super-aging society has changed the demographic structure of our society. People have longer lifespans than before, and these changes are expected to accelerate further with the development of the medical industry [34]. The Republic of Korea is the country with the fastest change in demographic structure, showing the lowest fertility rate in the world in addition to an aging population due to life extension [35]. As various factors cause these social changes, social responsibilities have emerged, one of which is to encourage physical activity for healthy aging. In this respect, this study aimed to activate the participation of older adults in leisure sports. Specifically, by analyzing the leisure sports participation constraints experienced by different age groups, this study discovered and analyzed participation constraints affecting older adults. Additionally, this study analyzed and compared psychological price sensitivity of different age groups in participating in leisure sports. This is essential at a time when the concept of leisure is changing into one of consumption [26]. In other words, this study verified the essential costs of leisure participation by age group in psychological terms to prepare a plan for older adults who have retired from economic activities to more actively participate in leisure sports. The results and interpretations of older adults’ participation in leisure sports analyzed in various aspects of this study are as follows.

First, the factors that did not show statistically significant differences in the three age groups examined in this study were health and social factors. These are sub-factors of leisure sports participation constraints, which have been considered the main limitations on people’s participation in leisure sports and have been analyzed in various existing studies [36,37]. The health factor refers to a situation in which people experience physical difficulties in participating in leisure sports [38]. In general, it could be easily regarded as a participation constraint factor only experienced by older adults; however, it showed no statistically significant group difference by age in this study. The emergence of new types of sports, which have recently modified existing sports, may have an impact. For example, “pickleball”, which is modified tennis or table tennis, can be enjoyed by men and women of all ages [39]. Additionally, park golf, a miniature version of normal golf, has been in the spotlight as a sports event that older adults can choose while minimizing physical burden [40]. These changes in sports may be an opportunity for older adults who may experience a decrease in their physical ability to more actively participate in leisure sports [41]. Of course, the main reason for this could be improvements in the physical ability of older adults due to the development of medical technology, which may have minimized group differences.

Next, the social factor did not show statistically significant group differences in this study. As humans are social animals, relationships with people around us might have an important effect on participating in leisure sports. Social benefits can be obtained through participation in leisure sports [42,43]; however, older adults could experience social isolation owing to changes in social roles after retirement [44]. Although it can provide social benefits, participation in leisure sports can act as a constraint for older adults experiencing social isolation. Nevertheless, this study confirmed that there was no difference in the constraints relating to social relations faced by older adults compared to the other two age groups; thus, the increase in the older adult population may have played a role, as a super-aging society has recently emerged. Additionally, the system’s improvement at the government level for the older adult population has had an impact. From a social perspective, the direction of the life sports policy for the older adult population in recent years has shifted from one of health promotion to welfare [45]. This is because it has been established that the most effective way to ensure older adults’ welfare in a super-aged society is to achieve physical, mental, and social health through physical activity. These government policies may solve social isolation through public sports facilities in each region and encouraging participation in leisure sports.

Third, the cost factor showed a significant difference by age group in the way it constrained participation in leisure sports. The concept of leisure, which in the past meant an activity for rest in surplus time (excluding essential working hours), has recently changed. Owing to the shortening of working hours and material abundance, the concept of leisure has changed from rest to consumption [9]. Sports with high participation costs have recently become more popular. In other words, participation in leisure sports through purchasing activities mirrors the purchasing of products. In this situation, previous studies have shown that older adults who have to consider retirement after being actively engaged in economic activities may be more vulnerable than the younger age group [46]. However, this study obtained the opposite result: the economic burden of participation costs was lower in the older age group than in the two relatively young groups. This seems to align with the current situation, in which economic growth has been achieved. These results might be considered positive results for the activation of older adults’ participation in leisure sports. It can be argued that it may be effective to propose a wider variety of quality leisure sports programs without any financial burden to older adults.

Finally, among the leisure sports participation constraints analyzed in this study, the time factor also showed statistically significant differences between age groups. Among the three participation constraints proposed by Crawford and Godbey [13]—(a) psychological states and attributes of independents, (b) interpersonal constructs, and (c) structural constructs—the time factor was the most representative of structural constraints, which were the strongest psychological influence on individuals. This means that people did not have time to participate in leisure sports owing to their busy daily lives. Older adults could be more at liberty to participate than those in the younger age group, who actively engage in economic activities. However, because social changes such as retirement do not guarantee time, our result could be linked to the cost factor mentioned above. This is because the economic margin of older adults can lead to a time margin after retirement. In the end, unlike older adults who were regarded as vulnerable groups in the leisure sports industry in the past, this study argues that older adults are the age group that is the most free from various factors restricting people’s participation in leisure sports. To summarize the above results, with regard to health and social relations, older adults are not more vulnerable than the relatively young age group and are rather more wealthy in both money and time. This oppositional result is derived from factors traditionally considered to make older adults vulnerable. Just as older adults are rapidly entering a super-aging society, their social situation is also rapidly changing.

Additionally, this study adopted the PSM technique, using age group to determine what people think the actual cost of participating in leisure sports might be. As confirmed by analysis of the cost factor of leisure in this study, older adults had more economic leeway than the relatively younger age groups. However, it is essential to analyze specific costs that are incurred by participation in leisure sports, and this is in line with consumer consumption activities. When interpreting the results of this study, this study found the most notable part to be the acceptable price range. Specifically, three groups showed distinctly different acceptable price ranges: (a) for the younger age group, 100–200 USD; (b) for the middle age group, 150–300 USD; and (c) for the older age group, 250–500 USD. Moreover, the smaller the difference between the results of the PME and PMC perceived by consumers, the greater the resistance to price. Our results found that the result value of the older age group was 250 USD, which was greater than that of the other two groups (younger age group: 100 and middle age group: 150). This means that older adults were relatively less sensitive to price fluctuations and more likely to participate in leisure sports even at higher costs. Finally, the reasonable price perceived by the consumer can be determined according to the difference between the values of OPP and IPP. As shown by this study, if the difference between OPP and IPP in all three age groups (younger age group: −40, middle age group: −50, and old age group: −40) is negative, it is possible to present a price close to the PME of consumers. Therefore, if interpreted based on our results, the average monthly price of participating in leisure sports that could be presented to the three age groups was 200 USD (younger age group), 300 USD (middle age group), and 500 USD (old age group), respectively. This study found that the appropriate price for older adults to participate in leisure sports was higher than that for other age groups and was not sensitive to price fluctuations. Based on these results, constructing a leisure sports program for older adults will be effective in revitalizing participation.

## 5. Conclusions

Social changes, such as technological advances and improvements in living environments, have extended the human lifespan. Consequently, the demographic structure of society has changed, and the average age of the population has increased. As the age of society members has increased, attention to health has also increased. Among the various methods to sustain healthy aging, the argument that the method of participating in regular physical activities is effective could be persuasive. However, because various factors limit one’s efforts to participate in leisure sports, it is essential to compare and analyze different experiences of leisure sports constraints by age to revitalize older adults’ participation in leisure sports. Additionally, the realistic cost of participating in leisure sports has limited people’s participation for a long time and has been regarded as an important factor in conjunction with people’s recent consumption activities. In this context, this study (a) compared individuals’ leisure sports participation constraints by age, (b) analyzed participation price sensitivity, and (c) made meaningful results and suggestions. This study could contribute to the health of older adults within the aging social structure by encouraging their participation in physical activities, which has undergone rapid changes in recent years. Nevertheless, the research process had some limitations; thus, suggestions for further research are as follows.

First, in this study, we applied quantitative research through questionnaires to a research design based on voluntary participation. By recruiting more than 300 survey participants, we increased the validity and reliability of the research data to derive objective and meaningful results for revitalizing older adults’ participation in leisure sports. However, research results obtained through quantitative research may have limitations in individually analyzing people’s diverse opinions. Therefore, further research to be conducted in the future should draw more diverse analysis results through in-depth interviews with individual research participants. Next, this study adopted the price of participation as the most important factor, focusing on the meaning of leisure, which has changed from a concept of relaxation to one of consumption. We presented a more realistic price through PSM, which has been considered effective in activating older adults’ participation in leisure sports. However, because people’s consumption behavior tends to be complex and anomalous, it is necessary to periodically analyze factors other than cost or price. For example, interest in additional factors such as residential areas, lifestyles, and housing types that were not included in this study will also have to be addressed in future studies. Finally, this study compared groups subdivided by age group with the aim of revitalizing leisure sports for older adults. As a result, this study examined the tendency of the relatively older age group to participate in leisure sports. However, additional research should adopt a research design limited to the age of 65 or older for research participants. More added variables may allow us to examine more specific behaviors within leisure sports participation among older adult participants.

## Figures and Tables

**Figure 1 behavsci-14-00803-f001:**
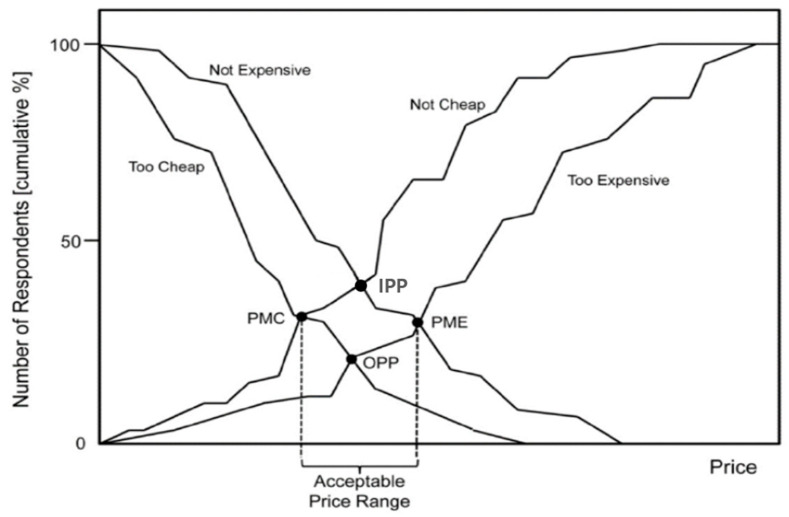
Price sensitivity meter from van Westendorp [16].

**Figure 2 behavsci-14-00803-f002:**
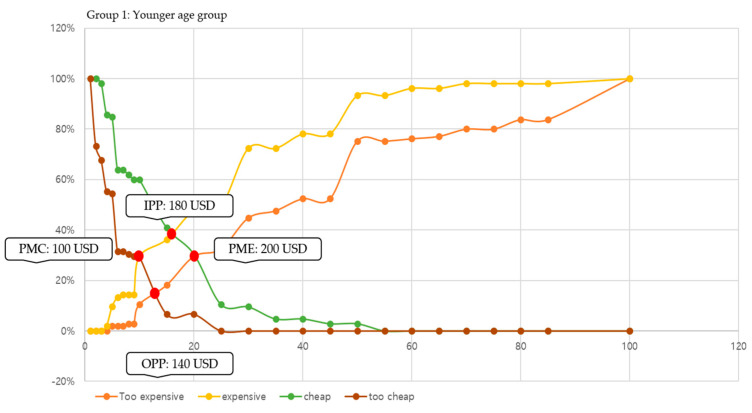
Result of PSM in the younger age group (Group 1).

**Figure 3 behavsci-14-00803-f003:**
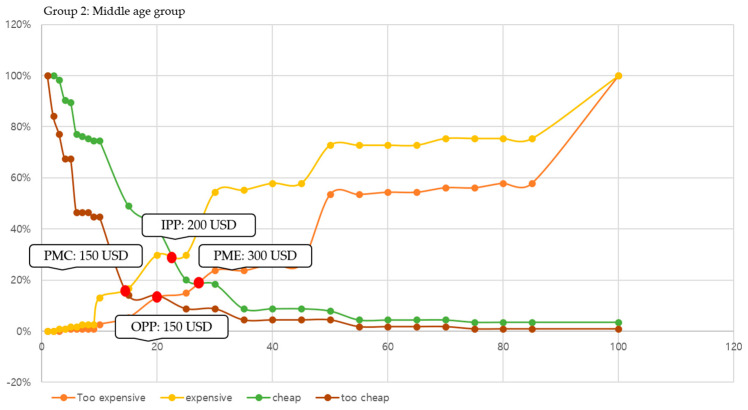
Result of PSM in the younger age group (Group 2).

**Figure 4 behavsci-14-00803-f004:**
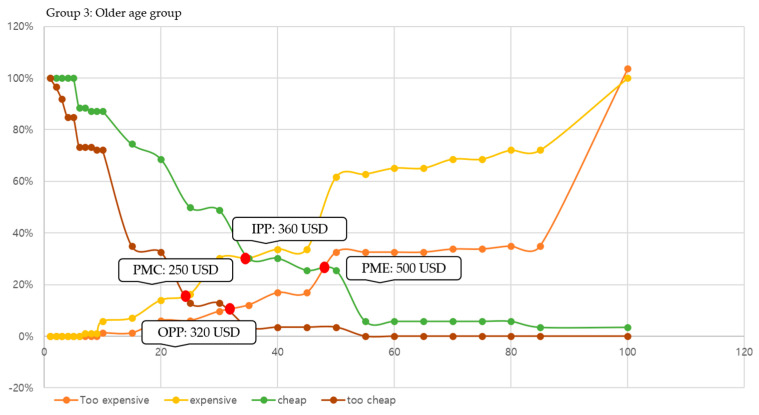
Result of PSM in the younger age group (Group 3).

**Table 1 behavsci-14-00803-t001:** Sociodemographic information of survey respondents.

Participant Characteristics	Subcategories	Group 1 Younger Age (*n* = 105)	Group 2 Middle Age (*n* = 114)	Group 3 Older Age (*n* = 86)
Sex	Male	40 (38.1%)	66 (57.9%)	42 (48.8%)
	Female	65 (61.9%)	48 (42.1%)	44 (51.2%)
Average monthly income	Less than 1000 USD	45 (42.9%)	5 (4.4%)	7 (8.1%)
	1000 USD–3000 USD	50 (47.6%)	16 (14.0%)	14 (16.3%)
	3000 USD–5000 USD	8 (7.6%)	37 (32.5%)	18 (20.9%)
	5000 USD–7000 USD	1 (1.0%)	15 (13.2%)	21 (24.4%)
	More than 7000 USD	1 (1.0%)	41 (36.0%)	26 (30.2%)
Time participating in leisure sports (years)	Less than 5 years	44 (41.9%)	29 (25.4%)	6 (7.0%)
	5–less than 10 years	32 (30.5%)	14 (12.3%)	10 (116%)
	10–less than 15 years	26 (24.8%)	15 (13.2%)	5 (5.8%)
	15–less than 20 years	2 (1.9%)	16 (14.0%)	13 (15.1%)
	Over 20 years	1 (1.0%)	40 (35.1%)	52 (60.5%)
Frequency of participation in leisure sports (per week)	Less than a day	27 (25.7%)	43 (37.7%)	12 (14.0%)
	1–2 days	33 (31.4%)	45 (39.5%)	39 (45.3%)
	3–4 days	34 (32.4%)	19 (16.7%)	17 (19.8%)
	More than 5 days	11 (10.5%)	7 (6.1%)	18 (20.9%)
Type of leisure sports	Individual	83 (79.0%)	98 (86.0%)	76 (88.4%)
	Team	22 (21.0%)	16 (14.0%)	10 (11.6%)
Total		105 (100.0%)	114 (100.0%)	86 (100.0%)

**Table 2 behavsci-14-00803-t002:** Results of scale validity and reliability.

Factors	Items	1	2	3	4
Social	My family/friends don’t want me to enjoy leisure sports	0.927	0.120	0.104	0.229
	I don’t have friends to enjoy leisure sports with	0.926	0.063	0.057	0.229
	My friends have interests other than leisure sports	0.898	0.080	0.102	0.151
Cost	I don’t have enough money to enjoy leisure sports	0.089	0.931	0.184	0.201
	Equipment for leisure sports is not reasonably priced	0.117	0.912	0.204	0.207
	The cost of leisure sports participation is too high	0.073	0.862	0.296	0.142
Time	The leisure sports take too long	0.086	0.257	0.894	0.136
	I don’t have enough time to participate in leisure sports	0.047	0.176	0.885	0.024
	It is hard to find the time to enjoy leisure sports	0.142	0.208	0.868	0.214
Health	I have too many health problems to participate in leisure sports	0.139	0.150	0.086	0.867
	I don’t have the energy to enjoy leisure sports	0.219	0.178	0.105	0.862
	I’m not fit enough to take part in leisure sports	0.295	0.206	0.178	0.732
	Eigenvalues	5.406	2.337	1.441	1.151
	Variance (%)	45.054	19.476	12.011	9.588
	Cronbach’s alpha	0.943	0.949	0.911	0.845

**Table 3 behavsci-14-00803-t003:** Results of multivariate analysis of variance on the factors of leisure participation constraints.

Factor	Sub-Factors	*df*	*F*	*p*	*η^2^*	G1	Mean G2	G3
Leisure Participation Constraints	Health	2	2.532	0.081	0.016	1.752	1.556	1.550
	Social	2	1.910	0.150	0.012	2.044	1.857	1.814
	Cost	2	24.904	0.000 ***	0.142	2.657	1.953	1.690
	Time	2	6.953	0.001 *	0.044	2.787	2.863	2.357

Note. *** *p* < 0.001, * *p* < 0.05.

**Table 4 behavsci-14-00803-t004:** Results of post hoc analyses between groups.

		Health	Social	Cost	Time
Group 1	Group 2	0.142	0.297	0.000 ***	0.857
	Group 3	0.169	0.205	0.000 ***	0.013 *
Group 2	Group 1	0.142	0.297	0.000 ***	0.857
	Group 3	0.999	0.945	0.182	0.002 *
Group 3	Group 1	0.169	0.205	0.000 ***	0.013 *
	Group 2	0.999	0.947	0.182	0.002 *
			-	a > b, c	a, b < c

Note. *** *p* < 0.001, * *p* < 0.05.

## Data Availability

The data applied in this research are available on request from the author.

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
