# Peer review of "A Study on the Leisure Sports Participation Behavior of the Elderly through Comparative Analyses by Age: Focusing on Leisure Participation Constraints and Price Sensitivity"

_behavsci, 2024, doi:10.3390/bs14090803_

Round 1

Reviewer 1 Report

Comments and Suggestions for Authors

Overall, interesting study. But, the title and paper do not match. Need to clarify your research population. The title suggests the research focuses on older adults, however in Line 13 the participants are over age 20, later you mention 3 age groups, with no specific age references. It appears that this is an age group comparison study not a study focused on older adults. However, literature examines older adults, not the three age groups.

Lines 99-101 are confusing c. and d. seem identical

Line 147-8 and 152 the population is divided into three age groups, younger age, middle age, and older age-but there is no definition of the ages within these groups

Author Response

Reviewer 1

Overall, interesting study. But, the title and paper do not match. Need to clarify your research population. The title suggests the research focuses on older adults, however in Line 13 the participants are over age 20, later you mention 3 age groups, with no specific age references. It appears that this is an age group comparison study not a study focused on older adults. However, literature examines older adults, not the three age groups.

→ I totally understand your concern. First, the title has been revised to explain the subject of this study. In addition, I have added information about age group classification with references and why this comparative research design by age group (not just focus on older age group) could analyze the leisure sports participation of elderly.

Lines 99-101 are confusing c. and d. seem identical → Revised

Line 147-8 and 152 the population is divided into three age groups, younger age, middle age, and older age-but there is no definition of the ages within these groups → Added.

Reviewer 2 Report

Comments and Suggestions for Authors

There are some revisions that I would like the authors to address and/or to consider.

Abstract

1.      Please add sampling technique

2.      Please provide date of the study

Main Manuscript

1.      replace Ultra-aging society with “super-aging society”

2.      Please theoretically justify Hypothesis 1 and Hypothesis 2 in the introduction section

3.      How authors found that not been answered truthfully: excluding 14 that had not been answered truthfully

4.      How age was clustered  to Younger age Middle age Older age

5.      Amend it : demographic information ( “e.g.”) to “i.e.”

6.      Please describe how “leisure participation constraint measurement” was developed

7.      Please provide the study design and Sampling technique

8.      How data collection was conducted

9.      Replace freer  with  more free

10.   Has the study conducted only by one researcher?

11.   Conflicts of Interest  section, it has been stated authors “The authors declare no conflicts of interest”

Author Response

Reviewer 2

There are some revisions that I would like the authors to address and/or to consider.

Abstract

  1. Please add sampling technique

→ Added (Based on quantitative research design using survey method with a convenience sampling technique, The 305 collected survey responses were analyzed for validity, reliability, and exploratory factor analysis through SPSS 28.)

  1. Please provide date of the study

→ Added From May 22 to July 10, 2024 in three community and sports centers, Korean adults over the age of 20 who regularly participated in leisure sports voluntarily responded to a questionnaire.

Main Manuscript

  1. Replace Ultra-aging society with “super-aging society” → Revised
  2. Please theoretically justify Hypothesis 1 and Hypothesis 2 in the introduction section I have added why this comparative research design by age group (not just focus on older age group) could analyze the leisure sports participation of elderly to the justify Hypothesis 1 and Hypothesis 2.
  3. How authors found that not been answered truthfully: excluding 14 that had not been answered truthfully → Revised (excluding 14 incomplete questionnaires)
  4. How age was clustered to Younger age Middle age Older age → I have added information about age group classification with references.
  5. Amend it: demographic information (“e.g.”) to “i.e.” → Revised
  6. Please describe how “leisure participation constraint measurement” was developed → Revised (The concept of leisure constraints has been strengthen with references.
  1. Please provide the study design and Sampling technique → Revised
  2. How data collection was conducted → Revised
  3. Replace freer with more free → Revised
  4. Has the study conducted only by one researcher? → Yes, this study has been conducted by one researcher.
  5. Conflicts of Interest section, it has been stated authors “The authors declare no conflicts of interest” → Revised

Round 2

Reviewer 2 Report

Comments and Suggestions for Authors

I would like to thank the author for considering the comments and changing the manuscript accordingly. 

Author Response

I would like to thank the author for considering the comments and changing the manuscript accordingly. 

 - Thank you for your support.